# Peer review of "Strategies for Integrating Generative AI into Higher Education: Navigating Challenges and Leveraging Opportunities"

_education, doi:10.3390/educsci14050503_

Round 1

Reviewer 1 Report

Comments and Suggestions for Authors

Author Response

Dear reviewer

We want to extend our sincere thanks for your thorough review of our manuscript. We recognize the importance and relevance of your comments and believe they are instrumental in refining our work. Please see the table below with the details of the revisions needed to the manuscript and our response.

Please see the attached revised manuscript (The changes are marked in yellow)

Kind Regards, The authors

Reviewer no 1

Comment

Response/changes made

Quality of English Language

English language fine. No issues detected

The manuscript has been professionally proofread/edited.

Is the content succinctly described and contextualized with respect to previous and present theoretical background and empirical research (if applicable) on the topic? - yes

Are all the cited references relevant to the research? yes

Are the research design, questions, hypotheses and methods clearly stated? Must be improved

It is not empirical research in terms of research questions, research instruments, or data collection and analysis. Please see the clarifications below.

Are the arguments and discussion of findings coherent, balanced and compelling? Must be improved

 There are no "findings" for this paper. It is not empirical research in terms of research questions, research instruments, or data collection and analysis. Please see clarifications below.

Is the article adequately referenced? yes

Are the conclusions thoroughly supported by the results presented in the article or referenced in secondary literature? Yes

Manuscript Review

1.

Originality:

Low.

The literature of recent years has generated a considerable number of publications focused on the theoretical, ethical, and organizational aspects of artificial intelligence in the educational field. This research segment is beginning to be inflated, at the expense of practical and concrete experiences regarding the use of these new technologies. Therefore, the originality of this manuscript is modest. It discusses the profound impact GenAI could have on learning and teaching practices, highlighting concerns such as inaccuracy, bias, overreliance on technology, and limited access to educational AI resources. There is already a significant body of literature based on similar topics and this contribution does not introduce any novelty.

Authors response

We added literature on AI (see lines 89-98)

Before the GenAI breakthrough, a range of studies explored the opportunities, challenges, and future research recommendations related to AI in education. For instance, Tahiru [21] and Zawacki-Richter [22] highlighted the potential role of AI in education, particularly in academic support and institutional and administrative services. However, they noted the challenges and risks associated with AI, such as the lack of critical reflection and the need to explore ethical and educational approaches. Further, Ali [23] and Okonkwo [24] delved into specific applications of AI in HE, including intelligent tutoring systems and chatbots. They emphasized the need for additional exploration of pedagogical, ethical, social, cultural, and economic dimensions and the potential benefits and challenges of these applications.

2.

Scientific Quality:

Weak.

The manuscript might be classified as a literature review. However, the content focuses more on a speculative approach towards regulatory aspects of AI usage, on ethical issues and the changes that the new technology can bring about.  We do not find a scientific methodology, not even in the

literature review framework; rather, an informative approach is adopted.

Authors response

The study is not a literature review, as might be understood from the manuscript title. We updated the title to be more apparent that it is a paper that maps the top considerations for implementing GenAI in higher education systems:

Strategies for Integrating Generative AI into Higher Education: Navigating Challenges and Leveraging Opportunities

3.

Relevance to the Field(s) of this Journal:

The paper fits the journal scope.

4.

Abstract:

The abstract aligns well with the content of the article. It introduces the topic of integrating Generative Artificial Intelligence (GenAI) in Higher Education (HE) and highlights the transformative potential and challenges associated with this adoption. It mentions the revolutionary capabilities of GenAI tools like ChatGPT, Midjourney, and Gemini, which are predicted to fundamentally transform various aspects of society, including education. The abstract also emphasizes the need for responsible integration of GenAI in HE to promote positive outcomes and suggests strategies for achieving this, such as creating awareness, training faculty, changing teaching

practices, partnering with students, promoting AI literacy, and conducting applied research. These themes and recommendations are consistent with   the broader discussion within the article regarding the implications, opportunities, and challenges of incorporating GenAI in higher education.

5.

Literature Review:

The engagement with the literature is comprehensive, encompassing a diverse range of pertinent perspectives, and the cited titles span from recent works to older ones, all of which are highly relevant to the study. However, we lack information regarding the method to select relevant publications.

Authors response

As might be understood from the manuscript's original title, the study is not a literature review. This is why we did not provide details on the method for publication selection. However, we did search for peer-reviewed academic papers relevant to the manuscript topic and included them (see  references list)

6.

Methodology:

Weak.

The methodology is based on literature review and a series of deductions stemming from authors’ considerations. But, accurate information regarding literature selection, adopted databases, and search criteria is not provided.

The contribution could have conducted a more in-depth analysis and critical evaluation of the existing studies around the practical use of GenAI in education, so, incorporating empirical evidence to support the recommendations and insights presented in the article.

Authors response

As might be understood from the original manuscript title, the study is not a literature review. This is why we did not provide details on the method for publication selection. However, we did search for peer-reviewed academic papers relevant to the manuscript topic and included them (see references list).

7.

Results:

Weak.

Some recommendations arising from the article, especially those in the 4th section - “Responsible Integration of GenAI in HE – Experts’ Recommendations” - echo the same patterns that have been proposed for two decades with the aim of updating teaching with well-known digital technologies. If we were to replace the acronym "GenAI" with, for example, ICTs, we would have content entirely compatible with an article written 15 years ago, before the advent of AI in education. Same approaches, same teaching strategies, same concepts, same recommendations that have paved the way for digital innovation.

Authors response

Indeed, the reviewer is right. There are similarities in every technological revolution that fundamentally changes learning and teaching processes. This is also true of the printing revolution. However, in our recommendations, we also referred to the unique characteristics of the current change upon the entry of the GenAI. For example, the need for learning literacies adapted to the GenAI era and students as co-partners with faculty.

8.

Discussions:

The manuscript does not feature a dedicated Discussion section as it serves as a regulatory document.

Authors response: please see our response below (to the conclusion comment)

9.

Conclusions:

There is no such section. The manuscript ends with a Summary.

Authors response

The paper is a collaborative work of academic experts trying to offer a preliminary outline of implementing GenAI in academic teaching. The paper has three parts: The first is a literature review (please note that we extended the references), which is still in its infinity. Based on the literature review, we offered our recommended strategies for integrating GenAI into HE to create the following positive outcomes: raise awareness about disruptive change, train faculty, change teaching and assessment practices, partner with students, impart AI learning literacies, bridge the digital divide, and conduct applied research. Finally, based on the first two parts, we propose four preliminary scale levels of a GenAI adoption for faculty. At each level, we suggest courses of action to facilitate progress to the next stage in adopting GenAI. We are aware of the manuscript's unique structure and hope it is suitable as a starting point for understanding the disruptive era we are in.

11.

References / Bibliography:

The engagement with bibliographic references is appropriate and includes also recent titles.

Reviewer 2 Report

Comments and Suggestions for Authors

Thank you for the opportunity to review this topical paper. I read it with interest.

On line 27, the authors cite a source from 2013 when discussing a ‘current’ phenomenon. I suggest revisiting this.

On line 33, the authors cite a non-peer-reviewed online article to support their argument. I advise against this.

On line 58, the phrase “…that has never been seen before” is redundant if referring to new (previously non-existing) data.

I recommend that the authors substantiate their claims on lines 71-88 and on lines 132-136 with existing literature. As it stands, these passages are nondescript.

On line 90, in line with the literature on learning design, the authors might consider the term “learning and teaching” instead of “teaching and learning”. This applies throughout the manuscript (it is used ‘correctly’, in my view, on line 108). It is a subtle but important difference.

On lines 90-96, the authors diminish the pedagogical process to a binary and sequential “transfer of knowledge”. This is a reductive view of pedagogy and warrants revision.

On lines 97-102 (and 104), the authors repeat the argument.

Concerning the remaining part of the manuscript, and especially Section 4 (‘expert recommendations’), I was left disappointed. The recommendations are firstly not discussed in conjunction with existing literature. I am here not referring to GenAI in higher education alone, but to digital technology in education. It is not a new field. GenAI certainly brings new ethical, social, political and other challenges, but the authors do little to contextualise their recommendations in a body of scholarship. Because of this, the recommendations lack rigour and substance.

Secondly, the proposed GenAI adoption model on page 7 is very interesting but again lacks substance. I would have liked to authors to delve into these stages, based not only on “expert recommendations” but also on empirical evidence and even theoretical discourse. As I mentioned before, digital technology in education is not new, and there are both empirical models and theoretical frameworks to understand adoption.

Taken as a whole, the paper does not contribute much to the scholarship on GenAI and education. It could be seen as somewhat of a summative paper, but that by itself does not merit publication. Significantly, the recommendations and proposed adoption model are written as general reflections and are not grounded (sufficiently or explicitly) in existing work.

Comments on the Quality of English Language

The paper can benefit from being professionally proofread/edited.

Author Response

Dear reviewer

We want to extend our sincere thanks for your thorough review of our manuscript. We recognize the importance and relevance of your comments and believe they are instrumental in refining our work. Please see the table below with the details of the revisions to the manuscript and our response.

Please see the attached revised manuscript (The changes are marked in yellow)

Kind Regards, The authors

Reviewer no 2

comment

Response/changes made

Quality of English Language

((x) Minor editing of English language required

The manuscript has been professionally proofread/edited.

Is the content succinctly described and contextualized with respect to previous and present theoretical background and empirical research (if applicable) on the topic? - Can be improved

We added literature on AI (see lines 89-98)

Before the GenAI breakthrough, a range of studies explored the opportunities, challenges, and future research recommendations related to AI in education. For instance, Tahiru [21] and Zawacki-Richter [22] highlighted the potential role of AI in education, particularly in academic support and institutional and administrative services. However, they noted the challenges and risks associated with AI, such as the lack of critical reflection and the need to explore ethical and educational approaches. Further, Ali [23] and Okonkwo [24] delved into specific applications of AI in HE, including intelligent tutoring systems and chatbots. They emphasized the need for additional exploration of pedagogical, ethical, social, cultural, and economic dimensions and the potential benefits and challenges of these applications.

Are all the cited references relevant to the research? Can be improved

We added references (highlighted in yellow in the reference list)

Are the research design, questions, hypotheses and methods clearly stated? Must be improved

It is not empirical research in terms of research questions, research instruments, or data collection and analysis. Please see clarifications below.

Are the arguments and discussion of findings coherent, balanced and compelling? Must be improved

There are no "findings" for this paper. It is not empirical research in terms of research questions, research instruments, or data collection and analysis. Please see clarifications below.

Is the article adequately referenced? Must be improved

Comments were corrected. Please see below.

Are the conclusions thoroughly supported by the results presented in the article or referenced in secondary literature? Can be improved

As mentioned, there are no "findings" for this paper. It is not empirical research in terms of research questions, research instruments, or data collection and analysis.

On line 27, the authors cite a source from 2013 when discussing a ‘current’ phenomenon. I suggest revisiting this.

Deleted

On line 33, the authors cite a non-peer-reviewed online article to support their argument. I advise against this.

changed to:

Carlopio, J. A history of social psychological reactions to new technology. J. Occup. Psychol., 1988, 61(1), 67–77.

On line 58, the phrase “…that has never been seen before” is redundant if referring to new (previously non-existing) data.

Deleted: that has never been seen before

I recommend that the authors substantiate their claims on lines 71-88 and on lines 132-136 with existing literature. As it stands, these passages are nondescript.

 Lines 71-88 changed the wording and added references (lines 61-73):

The widespread adoption of GenAI applications is attributed to several factors. Firstly, advancements in computing infrastructure, cloud computing, processing power, the availability of powerful algorithms, data harvesting, and storage have enabled the fast and mature handling of complex real-time AI models [13]. Additionally, the removal of technological barriers and the availability of simple language interfaces have made GenAI applications accessible and user-friendly [14]. The influence of technology-savvy student populations and the impact of social networks and globalization have also contributed to the readiness of individual users to explore GenAI applications [10]. However, there are concerns associated with the use of GenAI in education, particularly about the impact of GenAI on core skill development and the need for educators to guide students' interactions with GenAI to preserve these skills [15]. Lastly, the potential of AI-generated content to overcome limitations in traditional recommender systems has been highlighted, and there is a proposal to create a new generative recommender paradigm [16].

Lines 135-141 added references:

The literature shows that applications such as ChatGPT can be used to broaden knowledge and learning processes on diverse subjects [36–37], promote personalized learning experiences [38], and facilitate the writing process [39–40], thus reducing the pressure of writing papers [39] and cognitive load [36] while giving learners autonomy that will increase their sense of control over learning processes and even strengthen the sense of their ability [42] and allow them to receive constant and real-time feedback on their performance [43].

On line 90, in line with the literature on learning design, the authors might consider the term “learning and teaching” instead of “teaching and learning”. This applies throughout the manuscript (it is used ‘correctly’, in my view, on line 108). It is a subtle but important difference.

Corrected

On lines 90-96, the authors diminish the pedagogical process to a binary and sequential “transfer of knowledge”. This is a reductive view of pedagogy and warrants revision.

Corrected (lines 76-88):

Along with research, HE systems are dynamic ecosystems that foster not only the transfer but also the co-creation of knowledge through a multitude of pedagogical practices. The educational journey is characterized by an intricate interplay between teaching and learning, where faculty members are not merely knowledge transmitters but also facilitators of a learning environment that encourages critical thinking, problem-solving, and the development of competencies relevant to the present and future professional fields. These pedagogical strategies underscore the importance of active student engagement and self-education, leading to deep and meaningful learning outcomes [17].

On lines 97-102 (and 104), the authors repeat the argument

Corrected (lines 99-102) :

The intelligent capabilities of GenAI applications, such as ChatGPT, and their ease of use allow many opportunities for learning and diverse uses, offering great pedagogical potential. Therefore, it is believed that teaching, learning, and academic research will be greatly impacted by the introduction of GenAI [1–2, 25].

Concerning the remaining part of the manuscript, and especially Section 4 (‘expert recommendations’), I was left disappointed. The recommendations are firstly not discussed in conjunction with existing literature. I am here not referring to GenAI in higher education alone, but to digital technology in education. It is not a new field. GenAI certainly brings new ethical, social, political and other challenges, but the authors do little to contextualise their recommendations in a body of scholarship. Because of this, the recommendations lack rigour and substance.

We agree with the reviewer that GenAI is not a new field.

We added literature on AI (see lines 89-98)

Before the GenAI breakthrough, a range of studies explored the opportunities, challenges, and future research recommendations related to AI in education. For instance, Tahiru [21] and Zawacki-Richter [22] highlighted the potential role of AI in education, particularly in academic support and institutional and administrative services. However, they noted the challenges and risks associated with AI, such as the lack of critical reflection and the need to explore ethical and educational approaches. Further, Ali [23] and Okonkwo [24] delved into specific applications of AI in HE, including intelligent tutoring systems and chatbots. They emphasized the need for additional exploration of pedagogical, ethical, social, cultural, and economic dimensions and the potential benefits and challenges of these applications.

The manuscript focuses on the current disruptive era. We believe that extending the literature review to earlier technological revolutions is beyond the scope of this manuscript.

Also, please see our response to your  following comment.

Secondly, the proposed GenAI adoption model on page 7 is very interesting but again lacks substance. I would have liked to authors to delve into these stages, based not only on “expert recommendations” but also on empirical evidence and even theoretical discourse. As I mentioned before, digital technology in education is not new, and there are both empirical models and theoretical frameworks to understand adoption.

Taken as a whole, the paper does not contribute much to the scholarship on GenAI and education. It could be seen as somewhat of a summative paper, but that by itself does not merit publication. Significantly, the recommendations and proposed adoption model are written as general reflections and are not grounded (sufficiently or explicitly) in existing work.

The paper is a collaborative work of academic experts trying to offer a preliminary outline of implementing GenAI in academic teaching. The paper has three parts: The first is a literature review (please note that we extended the references), which is still in its infinity. Based on the literature review, we offered our recommended strategies for integrating GenAI into HE to create the following positive outcomes: raise awareness about disruptive change, train faculty, change teaching and assessment practices, partner with students, impart AI learning literacies, bridge the digital divide, and conduct applied research. Finally, based on the first two parts, we propose four preliminary scale levels of a GenAI adoption for faculty. At each level, we suggest courses of action to facilitate progress to the next stage in adopting GenAI. We are aware of the manuscript's unique structure and hope it is suitable as a starting point for understanding the disruptive era we are in.

The paper can benefit from being professionally proofread/edited.

The manuscript has been professionally proofread/edited.

Reviewer 3 Report

Comments and Suggestions for Authors

The document explores the role of Generative Artificial Intelligence in Higher Education, commencing with a succinct overview of GenAI, followed by a comprehensive review of studies on its current application in the teaching and learning processes within higher education. The initial sections of the paper thoroughly prepares what given in the second part, i.e, both some recommendations for the benefit of educational decision-makers, including academic staff, and the proposition of an adoption model of GenAI tailored for faculty, structured around four preliminary stages. The work appears to be highly coherent, comprehensive, and lucidly articulated, potentially serving as a source of inspiration for educators, faculties, and academic policymakers.

Author Response

Dear Reviewer,

We would like to express our heartfelt thanks for the time and effort you dedicated to reviewing our manuscript. We greatly appreciate your positive feedback, which encourages us to finalize the manuscript revision (please see the attached revised manuscript - The changes are marked in yellow). 

Kind Regards,  

The authors

Round 2

Reviewer 2 Report

Comments and Suggestions for Authors

The authors have addressed my concerns.

Author Response

Dear reviewer,

We want to extend our sincere thanks for your second review of our manuscript. Please see below your comments and our response with the details of the revisions we made to the manuscript

Kind Regards, The authors

  • Adding a clear description of the methodology: used databases, selected sources (N. and types), search criteria.

Authors response: Added (lines 104-108)

The search for research on GenAI in HE covered peer-reviewed journal articles published in English. The search queries “GenAI” OR “generative artificial intelligence” AND “higher education” were used to include papers with these terms in the titles, keywords, or abstracts published from December 1, 2022, to February 28, 2024. The search was executed mainly on Google Scholar, and 33 relevant papers were included in the literature review.

2) Making the results more specific to AI. Several findings can be easily applied to ICTs and digital technologies in general. Please stress the unique aspects that relate to AI.

Authors response: We have added explanations for the uniqueness of the current change in some of the sections, especially in section 3  -the changing teaching and assessment:

With the advent of the GenAI tools, a continuous pedagogical strategy shift is needed. A strategy that places students at the center of the educational experience, engaging them in authentic problem-solving activities, data analysis, decision-making processes, collaborative endeavors with peers, and producing concrete outputs that encapsulate their learning journey is required. (lines 241 – 246)

3) Adding a discussion and a conclusion sections.

             Authors response: Added discussion section (lines 346-372) and changed the summary section to a conclusion section:

The advent of GenAI in HE heralds a transformative potential to reshape pedagogical frameworks and institutional strategies. This paper has discussed the multifaceted nature of GenAI's integration into HE, highlighting both its transformative potential and the multifarious challenges it introduces, particularly in the realms of ethical considerations and teaching practices.

As the literature review shows GenAI offers profound opportunities to improve students’ learning experiences by facilitating learning environments tailored to students educational needs. These technologies enable the co-creation of knowledge, fostering an integrative and participative educational ecosystem. GenAI's ability to provide personalized learning experiences with constant and real-time feedback can significantly enrich educational outcomes and equip students with the necessary skills for future professional challenges.

However, the integration of GenAI also introduces major challenges. The potential for these systems to perpetuate existing biases and infringe upon privacy rights and unethical conduct poses significant concerns. HE institutions must adopt adequate strategies to address these challenges, ensuring that deploying GenAI technologies aligns with the foundational principles of academic integrity.

While integrating GenAI into HE presents remarkable opportunities for educational development, it also requires a nuanced understanding of its implications and a robust framework for its responsible use. By embracing a comprehensive approach emphasizing ethical integrity, pedagogical effectiveness, and continuous adaptation, HE institutions can navigate the complexities of GenAI integration. This can improve the quality of academic education and ensure that it remains relevant and responsive to the needs of current and future generations of students. Through thoughtful and strategic integration, GenAI can be leveraged to foster an enriched, inclusive, and forward-looking academic educational landscape.